# A reliable and inclusive method for assessing failure-causing mechanical wear in prosthetic feet

Michael A. Berthaume[1]*, Sikandar Qayyum[2], Ryley Ofoedu[3], Taosif Mohammad[4], Reza Hematti[5], Nana Akosua Asamoa-Bonsu[6], Someya Ali[6], Gemma Ranson[7]

1 Department of Engineering, King's College London, London, United Kingdom, 2 Coombe Sixth Form, New Malden, United Kingdom, 3 London Academy of Excellence Stratford, London, United Kingdom, 4 Eastbury Community School, Barking, United Kingdom, 5 Alperton Community School, Wembley, United Kingdom, 6 Preston Manor School, Wembley, United Kingdom, 7 Division of Mechanical Engineering and Design, London South Bank University, London, United Kingdom

* michael.berthaume@kcl.ac.uk

## Abstract

Wear assessment of prosthetic feet is critical for monitoring device performance and replacement, especially in rural or resource constrained areas where clinical follow-ups can be infrequent. Standardised, reliable methods for wear evaluation are limited. This study investigates the inter-observer reliability of a method for quantifying patterns and relative magnitudes of plantar surface wear of prosthetic feet and presents preliminary results for intra-observer error. Sixty-two plantar images of used SACH feet from Sri Lanka were scored by four independent groups a published method for quantifying prosthetic foot wear. A 20x10 grid was applied to each image, and wear was scored from 0–9 per cell based on visual indicators. Interobserver agreement was assessed on full (48 feet, 4 raters) and reduced (62 feet, 3 raters) datasets. Intraobserver reliability was tested by 2 raters re-scoring five images after 5＋days. Kendall's W evaluated agreement, and a modified scoring method was developed and evaluated. Interobserver agreement ranged from fair to excellent and was higher in the reduced dataset. Intraobserver agreement appears high, especially in central, less-worn areas of the foot. The modified method showed reduced intraobserver variability but lower interobserver agreement, largely due to increased ties affecting Kendall's W, not increased disagreement. The original and modified methods for wear qualification demonstrate high reliability, supporting their use as a low-cost, field-appropriate tool for monitoring prosthetic foot wear to enhance the evaluation and design of prosthetic feet, particularly in resource-limited settings. While the full method provides higher resolution data, the reduced method is more broadly applicable to different foot types. Clinically, this method offers a reliable, scalable approach to assessing prosthetic foot wear, potentially supporting community-based monitoring. Further testing and methodological alterations to encompass additional foot designs are required for clinical use.

**Data availability statement:** All relevant data are within the paper and its Supporting Information files.

**Funding:** We would like to thank In2STEM for providing students with the work placement and connecting the authors. We would further like to thank the JJCDR for help in collecting the feet. These organisations provided in-kind support by covering the salaries, overheads, etc., of employees who facilitated this research. Trips to Sri Lanka were funded in part by an NIHR PrOTeCTgrant (Award ID 16/137/45) award to Prof Anthony Bull (ImperialCollege London) and Research England funding awarded to MAB through London South Bank University (no award number). The funders had no role in study design, data collection and analysis, decision to publish, or preparation of the manuscript.

**Competing interests:** I have read the journal's policy and the authors of this manuscript have the following competing interests: MAB and GR work with the JJCDR, which collected the prosthetic feet, on prosthetic design, manufacture, and provision.

## Introduction

Prosthetics improve mobility, independence, and quality of life for those with lower limb loss [1–3]. While the socket transfers forces from the person to the prosthetic device, the foot transfers forces from the ground to the device, making it crucial for stability, mobility, and balance.

To meet mechanical safety requirements in high income countries (HICs), prosthetic feet must pass the International Organization of Standardization (ISO) 10328 or 22675 standards [4,5]. These standards test foot durability under static and dynamic loads, ensuring feet do not fracture or delaminate during use. While they work well for prosthetics in HIC, like the UK, they may not be appropriate for low- and middle-income countries (LMICs). For example, a study of 21 solid ankle cushion heel (SACH) and Jaipur prosthetic feet from various manufacturers/countries found feet often failed ISO 10328 tests despite being used safely in the countries of origin. This demonstrated the ISO test did not effectively predict foot failure and thereby safety in these LMICs [6].

### Prosthetic foot failure in LMICs: data from the field

There are two main styles of feet used in LMICs; the SACH and Jaipur feet. The majority of studies investigating SACH and Jaipur feet mechanics focus on their performance (e.g., [6–10]) and not on how they fail when they are being used.

Field data, which investigated feet used in LMICs, suggests SACH feet fail due to mechanical wear on the plantar surface, which is not considered in the ISO tests. An analysis of 66 prosthetic feet from Sri Lanka that were replaced as part of normal prosthesis maintenance were analysed to identify failure mode, and the primary cause of failure/replacement was determined to be wear. If fracture occurred, it was secondary to wear. Conversations with the Jaffna Jaipur Centre for Disability Rehabilitation (JJCDR) – the P&O clinic which manufactured and collected these feet – revealed wear was the primary factor used by the clinic to determine when feet needed to be replaced [11]. Similarly, 27 replaced SACH feet from the International Committee of the Red Cross's facility in Kinshasa, Democratic Republic of the Congo (DRC), were all replaced due to wear and not fracture (unpublished). ISO tests focused on fracture/delamination may therefore not be appropriate for LMICs.

Prosthetic foot wear on the plantar surface is common when prosthetic feet are donned without closed toe shoes, as is done in countries like Sri Lanka, Tanzania, Uganda, and the Democratic Republic of the Congo (MAB, personal observations). In Sri Lanka, shoes are often left at the door in homes and in public spaces, like university lecture halls, and people enter barefoot. Being barefoot or wearing open toed shoes, like sandals or crocs, allows abrasives (e.g., dirt or grit) to contact the plantar surface of the prosthetic foot, causing wear [11,12].

The failure of Jaipur feet was investigated in India, where 103 foot pieces across 98 individuals with limb loss were examined at the time of patient-reported (mechanical) damage. Location and severity of the damage was recorded, and it was found the interaction between a patient's total time spent standing and their total time spent wearing

the Jaipur foot, as well as the interaction between time spent standing and total distance walked with the Jaipur foot, were correlated with the location of damage to the prosthetic foot. The type of damage (wear vs fracture) was not recorded [13].

## Methods for quantifying prosthetic foot wear

Despite clinical relevance in LMICs and significant advances in tribology, to our knowledge, there is only one method for quantifying the mechanical wear of prosthetic feet [11]. Other methods for quantifying mechanical failure of prosthetic feet/ foot ankle components focus on fracture [13,14], fatigue [15–18], or temperature [15,16]. Often, fracture and fatigue are the main factors considered when testing prosthetic foot designs for LMICs [14,15,18] if mechanical failure is considered at all [19]. When wear is considered in prosthetic design, as is common in prosthetic orthopaedic devices [20–23], it is done so with joints as moving components are more susceptible to mechanical wear.

To bolster the methods used to investigate prosthetic foot wear, which appears to be the primary cause of foot failure for SACH feet, at least, in LMICs, we investigate inter- and intra-observer error rates in [11]'s method of foot wear quantification to test its reliability. We propose rules and suggestions for using this method, and alterations to the method for future analyses to increase reliability. We then re-interpret some results from [11] and suggest potential ways in which this method could be applied to other prosthetic feet in the future. Given wear is used by clinics in Sri Lanka and the DRC as a method for determining when prosthetic feet should be replaced, reliable methods for quantifying plantar wear on prosthetic feet and monitoring its progression are clinically relevant. We note that the method being investigated here is specific to SACH feet manufactured with nylon sheets on the plantar surface (see below), and therefore is not applicable to all prosthetic foot designs.

## Materials and methods

### Prosthetic foot sample

Between November 2019 and July 2022, 122 broken prosthetic feet were collected from the JJCDR, in Jaffna, Sri Lanka (Table 1 in [11]). Ethics were not required as feet were replaced as part of prosthetic maintenance, being disposed of, and foot users were unknown. Conversations with the prosthetists/technicians revealed all feet were replaced due to high levels of wear, particularly under the distal end of the keel, which can lead to fracture. Any failure (e.g., fracture) was believed to occur secondarily to wear. Feet were given accession IDs and are stored at King's College London.

**Table 1. Description of wear analysis score taken from [1]. Once a foot has been used, no cells can receive a score of "0". Scores of NA were given in previously, but its description was missing from the table.**

| Score | Description of wear |
| --- | --- |
| 0 | No wear (newly manufactured, not yet used) |
| 1 | No observable wear |
| 2 | First layer of nylon exposed |
| 3 | Second layer of nylon exposed |
| 4 | 0%–33% of rubber under nylon exposed |
| 5 | 34%–66% of rubber under nylon exposed |
| 6 | 67%–99% of rubber under nylon exposed |
| 7 | 100% of rubber under nylon exposed |
| 8 | Portions of rubber missing |
| 9 | Portions of foot completely missing |
| NA | Even if all portions of the foot were present, no portion of the foot would occupy this cell |

105/122 were JJCDR-manufactured derivatives of an older version of the International Committee of the Red Cross's SACH foot. To manufacture feet, a locally manufactured keel (polypropylene, injection moulded) is placed in an aluminium foot mould with two nylon sheets and surrounded by natural rubber. The nylon sheets are placed on the bottom of the mould to improve the foot's wear resistance and structural strength (see [11]). For 66 of these feet, the nylon was provided by a tire manufacturer on the island and reinforced with string, or cord; these feet, used in [11], were considered here.

## Scoring foot wear

Derived from dental macrowear studies [24], a 20 by 10 grid was placed over a photograph of the plantar surface of each foot and each square was qualitatively scored from 0 to 9, depending on the degree of wear (Table 1). Previously, it was found average wear was highest under the distal end of the keel and around the outer edge of the foot (Fig 5 in [11]).

## Validation of the wear scoring method

Methods for quantifying wear usually rely on quantifying, e.g., changes in mass or volume over time, which is useful for understanding how to predict wear over time. While useful for design and failure analyses, it is not needed here, where we are quantifying patterns and magnitude of wear on the plantar surface of the foot.

The design and manufacture of the feet used in this study is described in [11]. Briefly, two layers of fibre reinforced nylon sheets are placed on the bottom of a foot mould with natural rubber to create the plantar surface of the foot. Therefore, the exposure of the first, then the second, layer of nylon represents removal of material from the plantar surface, and the exposure of the second layer represents more material removal than the exposure of the first layer. This makes the method being investigated here specific to this design of prosthetic foot, and we have begun to investigate how to generalize this method for other designs of prosthetic feet (below).

## Reliability

To maximize the chances of interobserver error, and truly examine the reliability of this method, teenage students from local schools, who lacked backgrounds in prosthetics and therefore true novices (SQ, RO, TM, RH, NA, SA), were recruited for this study. The collaboration with novices is likely to introduce substantial variability, which likely mimics or exceeds the variability that would be observed in real-world (e.g., clinical) settings, where consistent training may be limited. These results therefore represent true "worst case" reliability scenarios and – if the method is reliable and robust in this setting – it will likely be reliable to be used in real-world settings.

As part of a public engagement project, seven students from local schools were recruited through the In2STEM charity to collect data as part of summer placements (12–23 August 2024). Images of the plantar surfaces of the prosthetic feet from [11] – taken using Motorola Moto G or a Nikon D7500 from a distance of at least 1 m – were provided to the students, along with the prosthetic feet. Right-foot images were mirrored to appear left.

## Training

**Day 1**: Groups (3 consisting of 2–3 students each) were given a picture of a worn prosthetic foot and, using PowerPoint, taught how to align, rotate, and scale the 20x10 grid to match foot length (toe to heel) and width (medial to lateral edges). When parts of the foot were missing, edges were estimated. Using a physical foot as reference, they scored each cell from 0 to 9 (Table 1), or NA if the foot was not present in the cell. Training lasted ~3–4 hours with formative feedback.

## Data collection

**Days 2–8:** Groups worked from a stack of randomly ordered labelled images. Generally, one student dictated scores while the other recorded scores. Feedback, but not scores, was provided when needed. Data were collected from 64 feet, as two were previously destroyed.

## Data checking

**Days 9–10:** Data was reviewed and it was discovered one group had not aligned the grid to the pictures of the plantar surfaces of the feet. This group recollected data for 48 feet. Other groups re-scored the first five feet they scored, each, enabling 5 + days between scores, for intraobserver analysis. A final review of data showed two feet were missing from 1 + dataset, reducing the interobserver sample to 62 feet.

## Statistical analysis

For the interobserver error analysis, groups were treated as single raters. Data from the 3 groups was combined with data from [11]. Kendall's coefficient of concordance (W) was used to assess interobserver agreement for each cell (i.e., "Do raters agree which feet are more/less worn for each cell?") and each foot (i.e., "Do raters agree which foot regions are more/less worn?"). Weighted Kendall's Ws were not used as published scores were from when the team was a novice with the method. Agreement was categorized as

- 0-0.20 poor/slight
- 0.21-0.40 fair
- 0.41-0.60 moderate
- 0.61-0.80 good/substantial
- 0.81-1.00 excellent

Two analyses were run: a reduced sample (4 raters, 48 feet) and a full sample (3 raters, 62 feet). Ties were corrected using a correction factor, and cells with NAs were excluded. Intraobserver error was calculated by counting cell-level agreement. Statistical analyses were performed in R v4.3.1 using Rstudio and the irr, stringr, dplyr, and raster packages [25–30].

## Artificial intelligence (AI)

Commonly employed AI tools, such as spell check, were used in the writing of this manuscript. ChatGPT was used to accelerate coding in R and to organise thoughts/streamline sections of the paper.

## Results

Interobserver reliability ranged from fair to excellent depending on the cell's location and number of observers (Fig 1). Higher agreement was generally observed near the centre of the foot and in areas of lower wear. The lowest agreement was observed at the proximal and distal edges of the foot. Variation in foot size (below) and grid placement meant cells around the bolt hole were often correctly assigned NA, and agreement could not be judged in these cells. The dataset with 3 raters had higher agreement than the dataset with 4 raters, potentially due to the fourth group having issues with the method, as was evidenced by errors made during data collection (see Data checking) (Fig 2).

Within feet, interobserver agreement was generally moderate to good/substantial with four raters and good/substantial with three raters, indicating raters generally agreed what parts of the feet were more/less worn. Cells near unworn areas of the foot or further from the foot's border tended to receive more consistent scores. Cells near high wear areas, or near the border of the foot, tended to have less agreement (Fig 3).

## Discussion

Mechanical failure of prosthetic feet results from both fracture and wear [11,12]. While fracture strength is tested using standardized methods (ISO 10328, ISO 22675), no equivalent standards exist for quantifying prosthetic foot wear. We tested the interobserver reliability and conducted a preliminary investigation into intraobserver reliability of a recently published, accessible method using novice school students to stress-test its robustness [11].

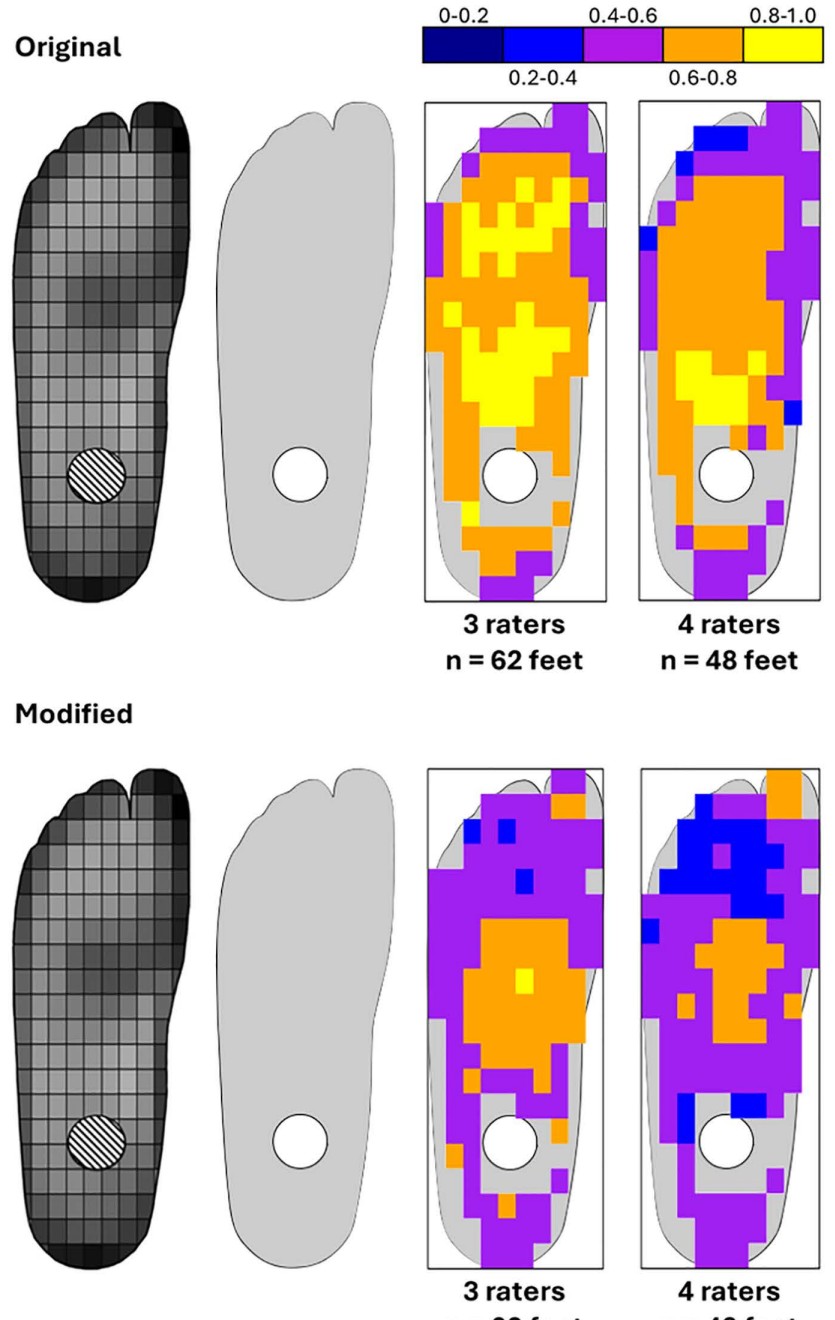

**Fig 1. Results from interobserver error analysis.** (left to right) Figure of average wear pattern on the plantar surface of the feet from [11]; a representative plantar prosthetic foot surface with bolt hole (white circle); visual representation of inter-observer reliability with binned Kendall's Ws. Visual representations of exact Kendall's Ws and test statistics provided in S1 Fig, S1 Table, and S2 Table.

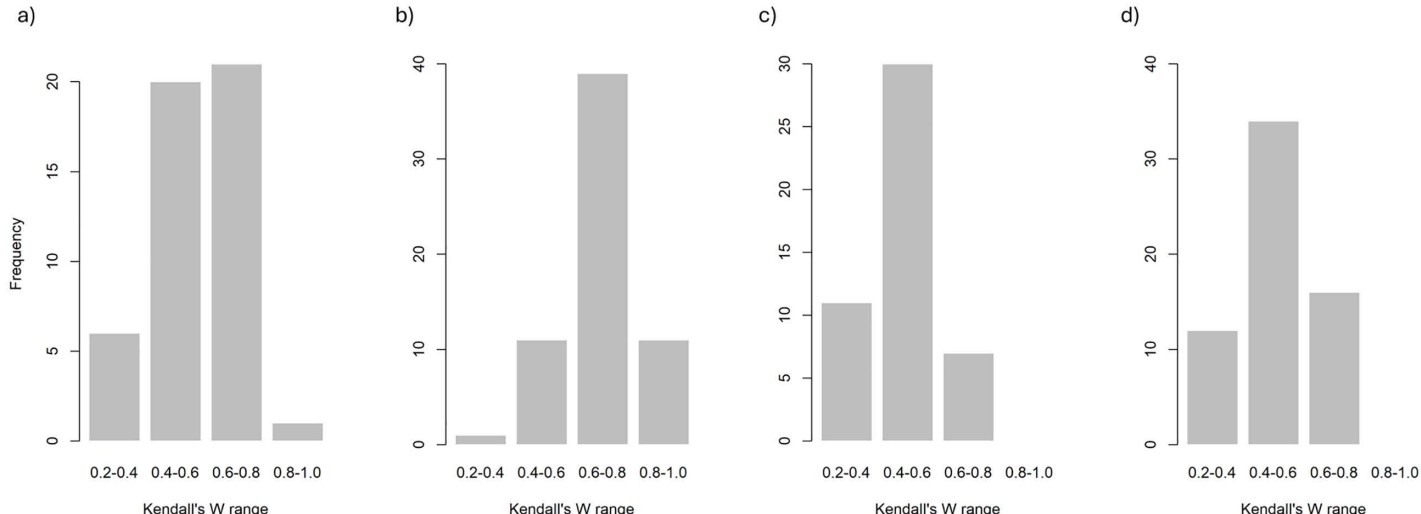

**Fig 2. Frequency plots of Kendall's Ws for interobserver error within feet for a) four and b) three raters using the method from [11], and c) four and d) three raters using the modified method.** Raw data in S3 Table.

Despite participants' inexperience, interobserver reliability was relatively high, especially in unworn regions, indicating the method's robustness. Agreement decreased along worn foot borders (Fig 1), likely due to difficulties in scoring or grid placement (Fig 4). Intraobserver reliability, while based on a small sample, also showed better consistency in central, unworn regions (Fig 3), but a larger sample is required to fully assess the intraobserver reliability of the method. As such, the results presented here should be taken as preliminary. The poorest interobserver agreement came from the fourth group, which failed to follow grid placement instructions. Their inconsistencies likely account for lower agreement in the four-rater analysis, not the method itself (e.g., LSBU-0137 cell R5C7 was given an 8, 8, and 9 by other groups but a 2 by this group; Table 1). However, the interobserver reliability of this method, despite issues with data collection, highlight its reliability for use in real-world settings.

### Issues identified with the published method and alterations

When collecting data, issues became apparent. During manufacture, nylon sheets did not reach the foot edges, meaning cells at the edge of the foot could not receive scores of 3, 4, 5, or 6, artificially elevating wear scores along the foot edges [11]. The published method is further not be applicable to nylon sheetless prosthetic feet, and therefore not generalizable to all designs of prosthetic feet. Issues the published method [11] can be categorized as follows:

1) **Judgement calls**: scoring cells on borders (edge effect, Fig 4a); estimating coverage (guesstimation, Table 1, scores 4–6); inconsistent grid placement leading to non-homologous cells (Fig 4b).

2) **Scoring method**: nylon is not always present in cells from time of manufacture; some cells can receive radically different scores (e.g., a 2 or 7)

3) **Feet**: all feet used the same keel, regardless of size, making the bolt hole and distal end of the keel relatively closer to the toes in smaller feet (Fig 3c), meaning cells in differently sized feet were not strictly homologous; some feet lack nylon sheets completely

4) **Time**: full scoring is laborious (200 cells/foot)

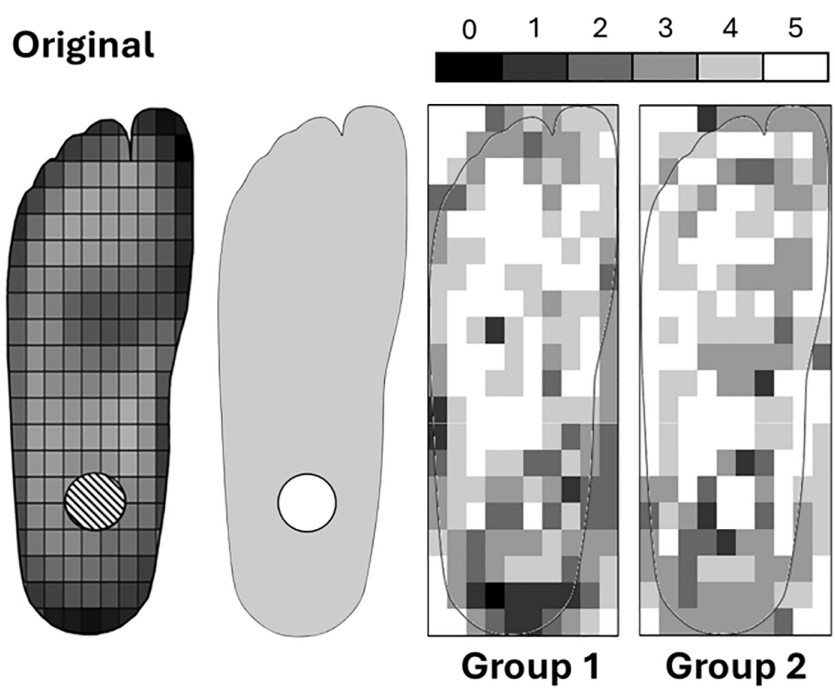

**Original**

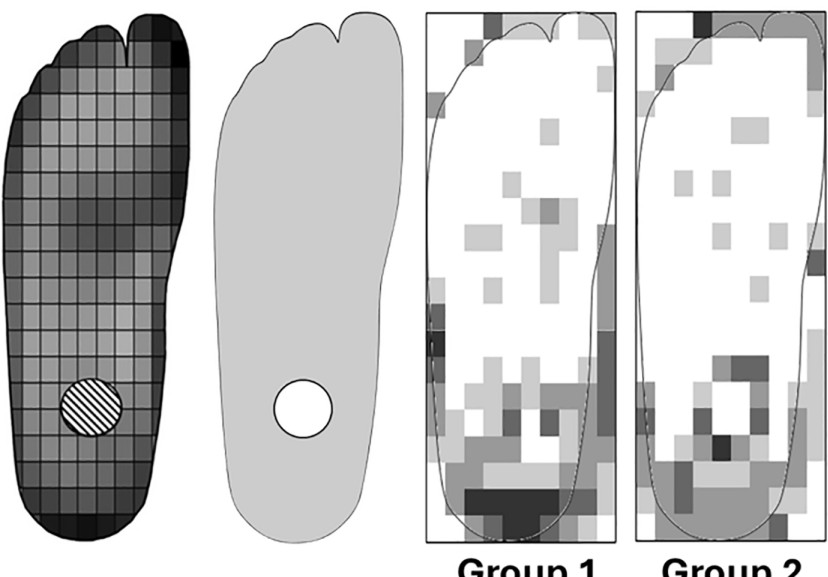

**Modified**

**Fig 3. Results from intraobserver error analysis, with scale representing the number of times cells received the same score.** Raw data in S1 Table.

PLOS Global Public Health

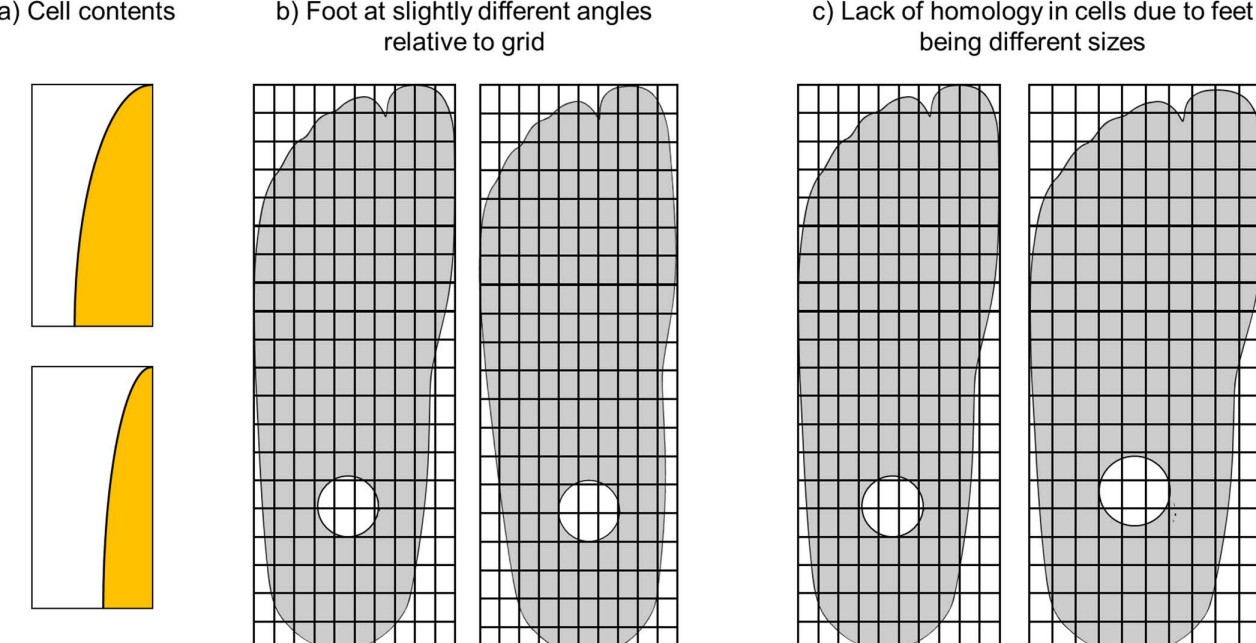

**Fig 4. Examples of some problems identified with the original version of the scoring method. a)** cells are partially occupied by the edge of the foot (orange), partially occupied by air (white), **b)** feet at slightly different angles relative to the grid, making cells not purely homologous, and **c)** the foot on the left is larger than the foot on the right, which is relatively wider. As they have the same size keel, the bolt hole is relatively closer to the heel in the smaller foot, and the cells containing the bolt hole are different.

To address these issues, we propose a set of rules and suggestions for scoring wear and modifications to the current method:

**Rules**

1) If cells are or should be 50%+filled by the prosthetic foot, they receive numerical scores, otherwise they receive "NA". E.g., Fig 4a upper image would receive a score but lower image an "NA".

2) If cells can receive multiple scores, record the highest score.

**Suggestions**

1) When guesstimating cell coverage (e.g., more or less than 50%), exact cell coverage can be quantified in image processing programmes like Adobe, Inkscape, or ImageJ.

2) Fitting methods can be employed with outlines of feet attached to grids to align grids to the bottom of the foot more reliably.

3) If comparing feet of multiple sizes, ensure cells are truly homologous or understand the average wear profile is not completely accurate and is dependent of foot size. In this case, take foot size into account during statistical analyses.

4) If time is an issue, grid size can be altered (e.g., 10x5) and/or only score wear in certain portions of the foot.

## Alterations

A minimalist version of the published method is proposed in Table 2. This method has broader applicability but lower resolution than Table 1.

## Reinterpretation of published results

Scores from Table 1 were collapsed into Table 2: scores from 0–1 remained as they were, 2–3 became 2, 4–7 became 3, 8 became 4, and 9 became 5. Though this is an imperfect translation, it enabled re-analysis of interobserver reliability using the simplified system.

Using this altered method, Kendall's W was generally lower (Fig 1, Fig 2) but this is not due to increased disagreement, but to increased ties introduced by the reduced number of possible scores. This narrows the range of the sum of squared rank deviations (S), lowering the numerator of the Kendall's W formula:

$$W = \frac{12S}{m^2 (n^3 - n) - mT} \tag{1}$$

Where m is the number of raters (4 or 3), n is the number of items being rated (48 or 62), and T is a correction factor for ties:

$$T = \sum_{k=1}^{g} \left( t_k^3 - t_k \right) \tag{2}$$

Where $t_k$ is the number of tied ranks in each score k of the g groups of ties [31]. Since m and n remained constant and it is impossible for there to be more disagreement in the modified vs. original system, T did not offset the drop in S and W decreased.

Intraobserver error was notably lower with the new method, with most of the cells receiving the same score (Fig 3). As before, edge cells had higher variability and these results should be taken as preliminary due to small sample size.

Overall, the modified method reduced score variability within individuals. While interobserver error is higher, this is related to how interobserver error is calculated and not variation between raters. Given the lower resolution in the modified method, we recommend the published method be used when possible, but the modified method (proposed here) be used when there are more raters and/or there is more variation in prosthetic foot design.

**Table 2. Proposed altered scoring method for quantifying wear on the plantar surface of prosthetic feet.**

| Score | Description of wear |
|---|---|
| 0 | No wear (newly manufactured, not yet used) |
| 1 | No observable wear |
| 2 | Wear present, but minimal (e.g., surface scratches, small gouges, or texture altered) |
| 3 | Wear present, and significant/advanced (e.g., deep scratches/gouges present or visible changes in foot height) |
| 4 | Portions of the foot are missing, beyond scratches/gouges, but the cell still contains portions of foot |
| 5 | Cell should be occupied by the foot but is not due to foot fracture or extreme wear |
| NA | Even if all portions of the foot were present, no portion of the foot would occupy this cell |

## Limitations

Using school students with no formal P&O training/knowledge increased variability in results. This was demonstrated clearly in one group, which stopped adhering to grid alignment instructions. Intraobserver analysis was also limited by time constraints, and data were checked on Day 8, which reduced interobserver error variability. The intraobserver analysis sample is small and results should be taken as preliminary.

The inclusion of variably sized feet with the same keel also introduced errors, as the area of high wear under the distal end of the keel was closer to the toes of smaller feet but the heel of larger feet. While the distance between the bolt hole and keel will remain constant [11], the position of the bolt hole relative to the toes will vary with foot size, being relatively closer to the toes in smaller feet.

The published method is ordinal and was constructed for a specific design of SACH foot. Higher scores indicate higher wear, but the intervals between scores are not uniform; this is somewhat exacerbated in the modified method. Designs of prosthetic feet that do not employ layers of nylon (e.g., foot shells) cannot directly apply this method.

Finally, this method is time consuming, but time for data collection can be reduced by creating larger and fewer cells.

## Conclusions

Fracture and wear both contribute to the mechanical failure of prosthetic feet, yet reliable, affordable methods for quantifying wear remain limited. In this study, we assessed the inter- and conducted a preliminary investigation into intraobserver reliability of a published method for quantifying patterns and magnitudes of prosthetic foot wear. We found the method to be generally robust, even when used by untrained raters. We identified key limitations in the original protocol and proposed a modified version with broader applicability. While the modified method offers lower resolution, it appeared to improve intraobserver consistency and is better suited for diverse foot types and larger rater groups. Together, these findings support the continued refinement of wear scoring tools to enhance the evaluation and design of prosthetic feet, particularly in resource-limited settings.

## Supporting information

**S1 Fig. 3 and 4 rater visualisation representations of exact Kendall's Ws and test statistics.**
(PDF)

**S1 Table. Raw data used in this study.**
(CSV)

**S2 Table. Raw Kendall's W scores.**
(CSV)

**S3 Table. Raw Kendall's W scores.**
(CSV)

## Acknowledgments

We would like to thank In2STEM for providing students with the work placement and connecting the authors. We would further like to thank the JJCDR for help in collecting the feet.

## Author contributions

**Conceptualization:** Michael A Berthaume.

**Data curation:** Michael A Berthaume, Someya Ali, Nana Akosua Asamoa-Bonsu, Reza Hematti, Taosif Mohammad, Ryley Ofoedu, Sikandar Qayyum, Gemma Ranson.

**Formal analysis:** Michael A Berthaume.

**Funding acquisition:** Michael A Berthaume.

**Investigation:** Michael A Berthaume, Someya Ali, Nana Akosua Asamoa-Bonsu, Reza Hematti, Taosif Mohammad, Ryley Ofoedu, Sikandar Qayyum, Gemma Ranson.

**Methodology:** Michael A Berthaume.

**Project administration:** Michael A Berthaume, Gemma Ranson.

**Resources:** Michael A Berthaume.

**Software:** Michael A Berthaume.

**Supervision:** Michael A Berthaume, Gemma Ranson.

**Visualization:** Michael A Berthaume.

**Writing – original draft:** Michael A Berthaume.

**Writing – review & editing:** Michael A Berthaume, Someya Ali, Nana Akosua Asamoa-Bonsu, Reza Hematti, Taosif Mohammad, Ryley Ofoedu, Sikandar Qayyum, Gemma Ranson.

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
