## [Decision Letter · Decision Letter 0]

12 Apr 2026

PGPH-D-26-00441

A reliable and inclusive method for assessing failure-causing mechanical wear in prosthetic feet

Dear Dr. Berthaume,

Thank you for submitting your manuscript to PLOS Global Public Health. After careful consideration, we feel that it has merit but does not fully meet PLOS Global Public Health’s publication criteria as it currently stands. Therefore, we invite you to submit a revised version of the manuscript that addresses the points raised during the review process.

We look forward to receiving your revised manuscript.

Kind regards,

Barnabas Tobi Alayande

Academic Editor

**Journal Requirements:**

i. Please clarify all sources of financial support for your study. List the grants, grant numbers, and organizations that funded your study, including funding received from your institution. Please note that suppliers of material support, including research materials, should be recognized in the Acknowledgements section rather than in the Financial Disclosure.

ii. State the initials, alongside each funding source, of each author to receive each grant. For example: "This work was supported by the National Institutes of Health (####### to AM; ###### to CJ) and the National Science Foundation (###### to AM)."

iii. State what role the funders took in the study. If the funders had no role in your study, please state: “The funders had no role in study design, data collection and analysis, decision to publish, or preparation of the manuscript.”

iv. If any authors received a salary from any of your funders, please state which authors and which funders.

2. Please send a completed 'Competing Interests' statement, including any COIs declared by your co-authors. If you have no competing interests to declare, please state "The authors have declared that no competing interests exist". Otherwise please declare all competing interests beginning with the statement "I have read the journal's policy and the authors of this manuscript have the following competing interests:"

3. Please note that your Data Availability Statement is currently missing the repository name and/or the DOI/accession number of each dataset OR a direct link to access each database. If your manuscript is accepted for publication, you will be asked to provide these details on a very short timeline. We therefore suggest that you provide this information now, though we will not hold up the peer review process if you are unable.

4. We noticed that you used “unpublished data" in the manuscript. We do not allow these references, as the PLOS data access policy requires that all data be either published with the manuscript or made available in a publicly accessible database. Please amend the supplementary material to include the referenced data or remove the references.

5. We have noticed that you have uploaded Supporting Information files, but you have not included a list of legends. Please add a full list of legends for your Supporting Information files after the references list.

**Additional Editor Comments (if provided):**

Thank you for this submission. The editorial team is in agreement with the documentation of a reviewer that this manuscript addresses quantifying wear-related failure in prosthetic feet, which is under explored in low-resource settings.

The manuscript submitted still has a few comments and unapplied references, giving the impression of a work in progress or unfinished version being submitted.

To reiterate very serious concerns raised, there are challenges seen in the following care aspects of the study that may prevent publication.

1) the study’s methodological rigor

a focus exclusively on inter- and intraobserver reliability, without establishing validity.

no comparison with any objective or quantitative measure of wear,

difficult to determine whether the scoring system reflects actual material degradation.

Study design- the use of untrained student raters is framed as a robustness test, but in practice, it introduces substantial variability and protocol deviations (including one group failing to correctly apply the method). This erodes the method’s usability and reproducibility.

intraobserver reliability is assessed using a very small subset (two raters, five images)- insufficient to draw strong conclusions.

reliance on Kendall’s W (which is sensitive to ties), the lack of complementary agreement metrics, and aggregating raters into groups, which may mask within-group variability.

The image-based scoring approach is not standardized (e.g., lighting, angle), which could further affect reproducibility.

2) the strength of its conclusions.

As a result of methodological loopholes, the conclusions, particularly those suggesting clinical applicability, appear overstated.

Challenges to External validity and generalizability (a single center in Sri Lanka, primarily of similar SACH-type construction), but claims are much more generalized.

Please reframe this as a preliminary reliability assessment rather than a fully validated method. While it has potential, substantial revision is needed to appropriately frame the contribution, temper the claims, and more clearly acknowledge limitations.

Kindly address each review comment in detail and apply changes to proceed.

Reviewers' comments:

Reviewer's Responses to Questions

**Comments to the Author**

1. Does this manuscript meet PLOS Global Public Health’s publication criteria? Is the manuscript technically sound, and do the data support the conclusions? The manuscript must describe methodologically and ethically rigorous research with conclusions that are appropriately drawn based on the data presented.

Reviewer #1: Partly

Reviewer #2: Partly

2. Has the statistical analysis been performed appropriately and rigorously?

Reviewer #1: I don't know

Reviewer #2: Yes

3. Have the authors made all data underlying the findings in their manuscript fully available (please refer to the Data Availability Statement at the start of the manuscript PDF file)?

Reviewer #1: No

Reviewer #2: Yes

4. Is the manuscript presented in an intelligible fashion and written in standard English?

Reviewer #1: No

Reviewer #2: Yes

5. Review Comments to the Author

**Reviewer #1:** Dear authors, the manuscript seems to be not finished, please resubmit when it is fully finished and revised. In the current form there are annotations and not completed references, before submitting everithing should be double checked.

**Reviewer #2:** This manuscript addresses an important and underexplored issue: quantifying wear-related failure in prosthetic feet, particularly in low-resource settings. The topic aligns with the journal’s scope, and the concept of a low-cost, field-applicable wear assessment tool is valuable. However, several concerns arise regarding the study’s methodological rigor and the strength of its conclusions.

The primary issue is that the manuscript focuses exclusively on inter- and intraobserver reliability, without establishing validity. There is no comparison with any objective or quantitative measure of wear, which makes it difficult to determine whether the scoring system reflects actual material degradation. Consequently, the conclusions, particularly those suggesting clinical applicability, appear overstated.

The study design also raises concerns. The use of untrained student raters is framed as a robustness test, but in practice, it introduces substantial variability and protocol deviations (including one group failing to correctly apply the method). This undermines confidence in the method’s usability and reproducibility, especially in real-world settings where consistent training may also be limited. Additionally, intraobserver reliability is assessed using a very small subset (two raters, five images), which is insufficient to draw strong conclusions.

External validity is another limitation. The dataset is restricted to prosthetic feet from a single center in Sri Lanka, primarily of similar SACH-type construction. This significantly limits the generalisability of the findings, yet the manuscript makes relatively broad claims about applicability across settings and prosthetic designs.

There are also statistical and methodological concerns, including reliance on Kendall’s W (which is sensitive to ties), the lack of complementary agreement metrics, and aggregating raters into groups, which may mask within-group variability. The image-based scoring approach is not standardized (e.g., lighting, angle), which could further affect reproducibility.

Overall, I view this study as a preliminary reliability assessment rather than a fully validated method. While it has potential, substantial revision is needed to appropriately frame the contribution, temper the claims, and more clearly acknowledge limitations.

6. PLOS authors have the option to publish the peer review history of their article (what does this mean?). If published, this will include your full peer review and any attached files.

**Do you want your identity to be public for this peer review?** For information about this choice, including consent withdrawal, please see our Privacy Policy.

Reviewer #1: No

Reviewer #2: **Yes:** Oluseun Adejugbe

**Figure Resubmissions:**

---

## [Decision Letter · Decision Letter 1]

12 May 2026

A reliable and inclusive method for assessing failure-causing mechanical wear in prosthetic feet

PGPH-D-26-00441R1

Dear Dr. Berthaume,

We are pleased to inform you that your manuscript 'A reliable and inclusive method for assessing failure-causing mechanical wear in prosthetic feet' has been provisionally accepted for publication in PLOS Global Public Health.

Best regards,

Barnabas Tobi Alayande

Academic Editor

All comments have been appropriately addressed. Thank you.

Reviewer Comments (if any, and for reference):

Reviewer's Responses to Questions

**Comments to the Author**

1. If the authors have adequately addressed your comments raised in a previous round of review and you feel that this manuscript is now acceptable for publication, you may indicate that here to bypass the “Comments to the Author” section, enter your conflict of interest statement in the “Confidential to Editor” section, and submit your "Accept" recommendation.

Reviewer #2: All comments have been addressed

2. Does this manuscript meet PLOS Global Public Health’s publication criteria? Is the manuscript technically sound, and do the data support the conclusions? The manuscript must describe methodologically and ethically rigorous research with conclusions that are appropriately drawn based on the data presented.

Reviewer #2: Yes

3. Has the statistical analysis been performed appropriately and rigorously?

Reviewer #2: Yes

4. Have the authors made all data underlying the findings in their manuscript fully available (please refer to the Data Availability Statement at the start of the manuscript PDF file)?

Reviewer #2: Yes

5. Is the manuscript presented in an intelligible fashion and written in standard English?

Reviewer #2: Yes

6. Review Comments to the Author

Reviewer #2: All the raised observation have been corrected

7. PLOS authors have the option to publish the peer review history of their article (what does this mean?). If published, this will include your full peer review and any attached files.

**Do you want your identity to be public for this peer review?** For information about this choice, including consent withdrawal, please see our Privacy Policy.

Reviewer #2: **Yes:** Oluseun Adejugbe
